# Leveraging Heterogeneous Side Information via Diffusion Models for Time-Series Anomaly Detection

## Abstract

In this paper, we propose a novel unsupervised learning approach for Out-of-Distribution (OoD) detection in time-series data, leveraging state-of-the-art diffusion models to capture the underlying data distribution. Our first contribution is the development of an effective OoD detector based on conditional sampling and reconstruction error measurement, eliminating the need for labeled data samples. We employ time series imputation techniques to reconstruct the original time series, enhancing the detection process. Our second contribution is the incorporation of domain-specific side information, which bolsters the diffusion model's ability to understand the structure of time-series data and results in a more robust OoD detector. Finally, we demonstrate the state-of-the-art performance of our proposed method through experiments on three diverse datasets: IoT Event Sequence Detection, DDoS Attack Detection, and Abnormal Network Transaction Sequence Detection. The experimental results highlight the effectiveness and versatility of our approach in addressing various OoD detection tasks across different domains.

## 1 Introduction

Out-of-distribution (OoD) detection is essential for determining if a given data point belongs to a specific domain, as machine learning models typically assume that test samples come from the same distribution as the training data. Deviations from the training distribution can lead to illogical results, which is especially important to avoid in high-stakes fields such as medicine, finance, and criminal justice. Previous studies have explored OoD detection in both supervised and unsupervised settings, with supervised scenarios benefitting from access to representative out-of-domain samples or in-domain data points with well-defined classes and ample class annotations.

A considerably more lenient assumption is to only require access to in-domain data points. Under this framework, methods have demonstrated competitive results for time series data. Although this approach is less informed, it relies on two implicit assumptions: that the in-domain data has well-defined classes and that there is an ample amount of data with class annotations. In reality, the condition of having labeled data is often not met. However, we usually have access to a substantial amount of unlabeled in-domain data. For instance, in the healthcare industry, electronic health records (EHRs) contain vast amounts of time series data, such as vital signs and lab results, but labeling this data can be time-consuming and expensive. Similarly, in finance, extensive historical market data is available, but labeling specific events or trends can be challenging and labor-intensive. Ideally, if we can accurately learn the distribution of in-domain data, an OoD detector should only need unlabeled in-domain data during the training process. As such, we formulate the problem of OoD detection as an unsupervised task, which aims to identify OoD samples without relying on labeled data or prior knowledge of out-of-domain samples.

In this paper, we propose that by utilizing the characteristic of the diffusion model, which learns a mapping to a manifold, we can develop a powerful unsupervised OoD detector for time series data. The underlying concept is that when a data point is elevated from its manifold, the diffusion model trained on the same manifold can restore the data point to its original vicinity. Conversely, if the diffusion model is trained on a distinct manifold, it will attempt to map the elevated data point toward its own training manifold, resulting in a significant distance between the original and

mapped data points. Consequently, we can identify out-of-domain time series data by examining this distance.

## 1.1 RELATED WORK

Out-of-distribution (OoD) detection research can be primarily classified into three categories: likelihood-based, feature-based, and reconstruction-based approaches. Likelihood-based approaches, first demonstrated by Bishop (1994), center around fitting the in-domain distribution using a specific model and evaluating the test data's likelihood under that model. Recent advancements often involve deep generative models, such as PixelCNN++(Salimans et al., 2017) and Glow(Kingma & Dhariwal, 2018), which support likelihood computation. However, studies by Choi et al. (2018), Nalisnick et al. (2018), and Kirichenko et al. (2020) have found that generative models sometimes assign higher likelihood to out-of-domain data than in-domain data. To tackle this issue, researchers have proposed strategies such as likelihood ratio approaches(Ren et al., 2019), adjusting likelihood based on data complexity and compression size(Serrà et al., 2019), optimizing model configuration(Xiao et al., 2020), typicality tests(Nalisnick et al., 2019; Morningstar et al., 2021; Bergamin et al., 2022), and improving generative model design choices(Maaløe et al., 2019; Kirichenko et al., 2020).

Feature-based approaches, on the other hand, focus on featurizing data in an unsupervised manner and fitting a simple OoD detector, such as a Gaussian Mixture Model, over the in-domain features. For example, Denouden et al. (2018) leverages autoencoder latent variables and evaluates the Mahalanobis distance in the latent space along with the data reconstruction error. Ahmadian & Lindsten (2021) extracts low-level features from the encoder of an invertible generative model, while Hendrycks et al. (2019); Bergman & Hoshen (2020); Tack et al. (2020); Sehwag et al. (2021) learn a representation over the in-domain data through self-supervised training. Furthermore, Xiao et al. (2021) demonstrates that a strong pretrained feature extractor can be used while maintaining comparable performance.

Reconstruction-based approaches, the category our approach falls into, assess how well a data point can be reconstructed by a model learned over the in-domain data. Within this line of work, several studies (Sakurada & Yairi, 2014; Xia et al., 2015; Zhou & Paffenroth, 2017; Zong et al., 2018) encode and decode data using autoencoders to evaluate the reconstruction quality. In contrast, Schlegl et al. (2017); Li et al. (2018) perform GAN(Goodfellow et al., 2014) inversion for a data point and evaluate its reconstruction error and discriminator confidence under the inverted latent variable. Concurrently with our work, Graham et al. (2022) employs diffusion models to reconstruct images at varied diffusion steps. Our approach, on the other hand, focuses on masking and inpainting an image repeatedly with fixed steps. The two approaches are complementary to each other and offer different perspectives on the problem of. Our work is most closely related to Liu et al. (2023), where authors deploy a similar method to ours but for image domain. Our work specifically focuses on timeseries domain.

## 1.2 CONTRIBUTIONS

As such our contributions are three-fold:

1. Our first contribution is the development of an unsupervised learning approach for OoD detection, leveraging state-of-the-art diffusion models to learn the data distribution. By utilizing conditional sampling for reconstruction and measuring the reconstruction error, we establish an effective OoD detector that does not require labeled data samples. Specifically, we use time series imputation techniques to reconstruct the original time series.
2. Our second contribution is the incorporation of domain-specific side information to improve OoD detection performance. By including side information in the learning process, our method enhances the diffusion model's ability to capture the underlying structure of time-series data, ultimately leading to a more robust OoD detector.
3. Finally, we experimentally demonstrate the state-of-the-art performance of our proposed method on three diverse datasets, encompassing a wide range of applications: IoT Event Sequence Detection, DDoS Attack Detection, and Network Beaconing Sequence Detection. These experimental results showcase the effectiveness and versatility of our approach in handling various OoD detection tasks across different domains.

## 2 PRELIMINARIES

### 2.1 DIFFUSION MODELS

Diffusion models (Sohl-Dickstein et al., 2015) represent a class of generative models that demonstrated state-of-the-art performance on a range of different data modalities, from image (Dhariwal & Nichol, 2021; Ho et al., 2020a; 2022a), over speech (Chen et al., 2020; Kong et al., 2021) to video data (Ho et al., 2022b; Yang et al., 2022).

On a high level, diffusion models sample from a distribution by reversing a gradual noising process. In particular, sampling starts with noise $x_T$ and produces gradually less-noisy samples $x_{T-1}, x_{T-2}, \ldots$ until reaching a final sample $x_0$. Diffusion models learn to remove the noise in a backward process that was added sequentially in a Markovian fashion during a so-called forward process. These two processes, therefore, represent the backbone of the diffusion model. For simplicity, we restrict ourselves to the unconditional case at the beginning of this section and discuss modifications for the conditional case further below. The forward process is parameterized as

$$q\left(x_1, \ldots, x_T \mid x_0\right) = \prod_{t=1}^{T} q\left(x_t \mid x_{t-1}\right) \tag{1}$$

where $q\left(x_t \mid x_{t-1}\right) = \mathcal{N}\left(\sqrt{1-\beta_t}x_{t-1}, \beta_t\mathbf{I}\right)[x_t]$ and the (fixed or learnable) forward-process variances $\beta_t$ adjust the noise level. Equivalently, $x_t$ can be expressed in closed form as $x_t = \sqrt{\alpha_t}x_0 + (1-\alpha_t)\epsilon$ for $\epsilon \sim \mathcal{N}(0, \mathbf{I})$ with $\alpha_t = \sum_{i=1}^{t}(1-\beta_i)$. The backward process is parameterized as

$$p_\theta\left(x_0, \ldots, x_{t-1} \mid x_T\right) = p\left(x_T\right) \prod_{t=1}^{T} p_\theta\left(x_{t-1} \mid x_t\right) \tag{2}$$

where $x_T \sim \mathcal{N}(0, \mathbf{I})$. Again, $p_\theta\left(x_{t-1} \mid x_t\right)$ is assumed as normal-distributed (with diagonal covariance matrix) with learnable parameters. Using a particular parametrization of $p_\theta\left(x_{t-1} \mid x_t\right)$, Ho et al. (2020a) showed that the reverse process can be trained using the following objective,

$$L = \min_\theta \mathbb{E}_{x_0 \sim \mathcal{D}, \epsilon \sim \mathcal{N}(0, \mathbf{I}), t \sim \mathcal{U}(1, T)} \left\| \epsilon - \epsilon_\theta\left(\sqrt{\alpha_t}x_0 + (1-\alpha_t)\epsilon, t\right) \right\|_2^2 \tag{3}$$

where $D$ refers to the data distribution and $\epsilon_\theta(x_t, t)$ is parameterized using a neural network, which is equivalent to earlier score-matching techniques (Song & Ermon, 2019; Song et al., 2021). This objective can be seen as a weighted variational bound on the negative log-likelihood that downweights the importance of terms at small $t$, i.e., at small noise levels.

Extending the unconditional diffusion process described so far, one can consider conditional variants where the backward process is conditioned on additional information, i.e. $\epsilon_\theta = \epsilon_\theta(x_t, t, c)$, where the precise nature of the conditioning information c depends on the application at hand and ranges from global to local information.

### 2.2 TIME SERIES IMPUTATION

Let $x$ be a data sample with shape $\mathbb{R}^{T \times F}$, where $T$ represents the number of time steps, and $F$ denotes the number of features or channels. Imputation targets are commonly defined using binary masks that correspond to the shape of the input data, i.e., $M \in \{0, 1\} T \times F$. In this case, ones indicate the values to be conditioned on, while zeros represent the values to be imputed.

Furthermore, we assume that each time series data point $x$ is accompanied by additional side information, denoted as $z \in \mathbb{R}^K$. The side information vector $z$ encapsulates supplementary details about the time series, incorporating domain-specific or auxiliary information related to the task. The side information could be quite heterogeneous in terms that they are collected from different sources or derived at different aggregation levels. For instance, if the time series represents stock market data, the side information could include the type of stock, historical volatility, and variance relative to other stocks, among other factors. The heterogeneity nature of side information usually makes it hard to get them integrated in time-series modelling.

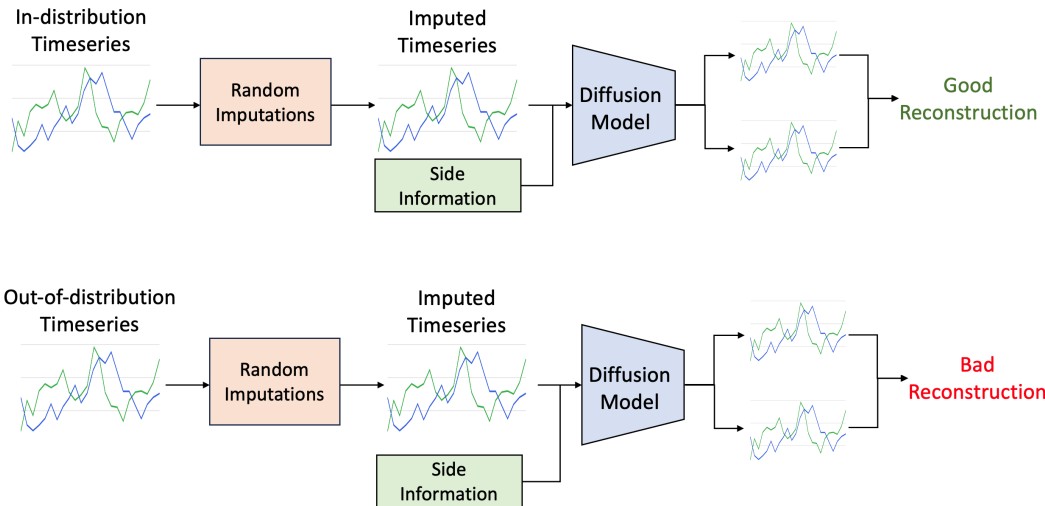

Figure 1: **Unsupervised Time Series OoD Detection Architecture:** This figure illustrates our method for detecting OoD time series data using a trained diffusion model to reconstruct in-domain data. The diffusion model takes imputed time series data $x$ and side information to reconstruct the original time series data. We compute the reconstruction error between the original and reconstructed time series. The top figure demonstrates a scenario where the test data point is in-distribution, resulting in accurate reconstruction. In contrast, the bottom figure depicts a scenario with an out-of-distribution test data point, leading to poor reconstruction. In summary, our architecture effectively distinguishes between in-distribution and out-of-distribution time series data based on the reconstruction quality.

## 3   PROBLEM STATEMENT

In this section, we define the problem of unsupervised OoD detection for timeseries data.

**Problem 3.1** (Unsupervised OoD Detection for Time-Series Domain)**.** Given a distribution of interest, $\mathcal{D}$, consisting time-series data, the objective is to formulate an unsupervised Out-of-Distribution (OoD) detection problem as follows: **Input**: A set of unlabeled in-distribution time-series samples with side information: $X = \{(x_1, z_1), \ldots, (x_n, z_n)\} \sim \mathcal{D}$, and a test data point $(x_{\text{Test}}, z_{\text{Test}})$. **Output**: An Out-of-Distribution score, $s(x_{\text{Test}}, z_{\text{Test}})$, that quantifies the likelihood of $x_{\text{Test}}$ not being sampled from $\mathcal{D}$.

Our goal is to construct a detector using the input set $X$, such that for a given test data point $s(x_{\text{Test}}, z_{\text{Test}})$, the detector computes an OoD score $s(x_{\text{Test}}, z_{\text{Test}})$, where a higher value of $s(x_{\text{Test}}, z_{\text{Test}})$ signifies a greater probability that x is not drawn from the distribution $\mathcal{D}$ in the timeseries domain.

This problem formulation captures many real-world scenarios of OoD detection problems, such as detecting anomalies in sensor data, identifying unusual patterns in financial time series, or detecting cyber security attacks. In these problems, we do not have OoD labels and only have access to in-distribution data.

## 4   OUR METHOD

In this section, we present our unsupervised Out-of-Distribution (OoD) detection algorithm for timeseries data. The objective of the algorithm is to compute an OoD score, $s(x_{\text{Test}}, z_{\text{Test}})$, for a given test data point $x_{\text{Test}}$ with side information $z_{\text{Test}}$. This score quantifies the likelihood of $x_{\text{Test}}$ not being sampled from the distribution of interest $\mathcal{D}$. Figure 1 shows an overview of our algorithm.

Our method is grounded in the understanding that an effective unsupervised OoD detector necessitates a large dataset representing the in-domain data and a model proficient in accurately learning the

data distribution. In numerous OoD detection problems, while a comprehensive in-domain dataset is often available, an OoD dataset may be lacking. Recently, diffusion models, a class of unsupervised generative models, have exhibited remarkable success in learning large-scale data distributions, making them well-suited for learning in-domain datasets. As unsupervised generative models, diffusion models do not require labeled data samples to learn the data distribution, enabling them to accurately model the in-domain data. Moreover, diffusion models are recognized for their precise conditional generation capabilities (Rombach et al., 2021; Ruiz et al., 2023). We capitalize on this ability to reconstruct in-domain samples by training the diffusion model on the given in-domain data, conditioned on imputed observed values, i.e. where we impute the observed time series data $x$ and condition the diffusion model to reconstruct it.

The key insight of our method lies in the fact that diffusion models excel at reconstructing in-domain data but struggle when it comes to reconstructing OoD data. Leveraging this insight, we develop an OoD detector based on the reconstruction error of the diffusion model when reconstructing data points conditioned on their imputed representations. By harnessing the reconstruction power of diffusion model, our proposed method is adept at addressing the challenges associated with unsupervised OoD detection for time-series data and constructing the OoD scoring function, $s(\cdot)$. The following subsections will walk through a few key advantageous design choices in our method.

## 4.1 Time-series Diffusion Model

In this work, since we are dealing with time-series data, we employ the Structured State Space Diffusion (SSSD) model (Alcaraz & Strodthoff, 2022). The SSSD model utilizes structured state space models as its internal model architecture (Gu et al., 2022), making it particularly suited to capture long-term dependencies in time-series data. As such, the SSSD model provides an ideal model architecture for our time-series dataset. We will train the SSSD model on the available in-domain dataset, $\mathcal{D}$, to learn the in-domain data distribution. The SSSD model learns to reconstruct the original timeseries, $x$, conditioned on its imputed representation $M \times x$, where $M$ is the imputation mask.

## 4.2 Incorporating Side Information

One of the key contributions of our method is the utilization of the side information vector $z$ along with the time-series data $x$. The side information encapsulates supplementary details about the time series, incorporating domain-specific or auxiliary information related to the task. By incorporating this side information into the algorithm, we can better understand the underlying structure of the time-series data and improve the ability of the DM to learn the dataset. For example, in the context of financial time-series data, the side information vector $z$ could include macroeconomic indicators, such as interest rates or inflation, or company-specific data, like earnings reports or news sentiment

## 4.3 OoD Scoring

For an input test time series $x_{\text{Test}}$ and side information $z_{\text{Test}}$, with its reconstruction $x_{\text{recon}}$ computed on an imputation mask, we can derive a OoD score in two ways. The first one is based on z-score[1]. Depending on the specific characteristics of the time series data, a distance can be computed between $x_{\text{Test}}$ and $x_{\text{recon}}$ (usually only on the masked time steps). A z-score describes the position of a raw value in terms of its distance from the mean when measured in standard deviation units. With a subset in-distribution time series, we can pre-compute the mean $\mu$ and standard deviation $\sigma$ of all the distances between time series and their imputations. Thefore we define our OoD score as $s = |\frac{\text{Distance}(x_{\text{Test}}, x_{\text{recon}}) - \mu}{\sigma}|$ and as a rule of thumb we use 3 as a threshold. On the other hand, when some OoD instances are also available, it's also possible to train a binary classifier on $(x_{\text{recon}} - x_{\text{Test}})$. In this case, the OoD score $s$ would correspond to probability of the in-distribution class from the classifier.

---

[1]https://en.wikipedia.org/wiki/Standard_score

**Input:** A test timeseries $(x_{\text{Test}}, z_{\text{Test}})$ and an SSSD model with parameters $\theta^\star$.
**Output:** anomaly score $s$

1: Generate random imputation mask $M$ for $x_{\text{Test}}$.
2: $\bar{x}_{\text{Test}} = M \times x_{\text{Test}}$
   **for** $t = T \; to \; 1$ **do**
3:     **if** $t == T$ **then**
4:        $x_{\text{recon}} \leftarrow$ sample from noise distribution.
     **end**
5:     $x_{\text{recon}} \leftarrow \text{SSSD}(x_{\text{recon}}, z_{\text{Test}}, \bar{x}_{\text{Test}}, \theta^\star)$
   **end**
6: Calculate OoD score $s \leftarrow |\frac{\text{Distance}(x_{\text{Test}}, x_{\text{recon}}) - \mu}{\sigma}|$ or via a binary classifier.
7: **Return** $s$.

**Algorithm 1:** The **Unsupervised OoD Detection Algorithm**

### 4.4 Algorithm Overview

Algorithm 1 shows the complete procedure for detecting OoD timeseries data. Our algorithm takes in as input a trained Structured State Space Diffusion (SSSD) model on the given in-domain dataset, $\mathcal{D}$. We will refer to this trained model as $SSSD(\cdot, \theta^\star)$, where $\theta^\star$ are weights of the final trained model. The input to the algorithm consists of a test time series $(x_{\text{Test}}, z_{\text{Test}})$, where $x_{\text{Test}}$ is the time series data point and $z_{\text{Test}}$ is the corresponding side information. The main steps of the algorithm are as follows:

1. Generate a random imputation mask $M$ for the test time series $x_{\text{Test}}$, and compute the masked time series $\bar{x}$Test by element-wise multiplication with $M$. Lines 1 and 2 in Alg. 1.
2. Start the reconstruction of the datapoint referred to as $x_{\text{recon}}$ when $t == T$, by sampling an initial random datapoint from a noise distribution a noise distribution. This is part of the general diffusion model sampling procedure. Lines 3 and 4 in Alg. 1.
3. Iteratively apply the SSSD with parameters $\theta^\star$ on $x_{\text{recon}}$, updating it at each step, from $t = T$ to 1. As stated before, the diffusion model reconstruction is conditioned on imputed timeseries $\bar{x}_{\text{Test}}$ and side information $z_{\text{Test}}$. Line 5 in Alg. 1.
4. Calculate the OoD score $s$ by computing the z-score $s = |\frac{\text{Distance}(x_{\text{Test}}, x_{\text{recon}}) - \mu}{\sigma}|$ or via a binary classifier (where $\mu$ and $\sigma$ are the mean and standard deviation of distances between a subset of in-distribution time series and their masked imputations).

The output of the algorithm is the OoD score $s$, which indicates the likelihood of the test data point $x_{\text{Test}}$ being out-of-distribution. A higher value of $s$ signifies a greater probability that $x_{\text{Test}}$ is not drawn from the distribution $\mathcal{D}$ in the time-series domain. Our algorithm's runtime is solely dependent on the sampling speed of the diffusion model.

## 5 Experiments

### 5.1 Datasets and Experiment Settings

#### 5.1.1 Datasets

We conducted experiments using two public benchmark datasets for time-series anomaly detection, and one production dataset shared by a prestigious cybersecurity vendor. Table 1 shows some statistics of the datasets.

The UNSW Internet of Things (IoT) Traffic Traces dataset [2] includes network transaction sequences from 31 devices (25 IoT and 6 non-IoT devices), recorded over a 60-day period. A sequence in the dataset is generated by one device. Due to computational limitations, our experiments use 20 days of transactions for all devices, spanning from September 23 to October 12, 2016. The dates are randomly selected. We process each transaction sequence into a multivariate time-series, following the methodology in Dou et al. (2020). Each time-series data point represents the transactions per

---

[2]https://iotanalytics.unsw.edu.au/iottraces.html

minute. One data point includes three categorical attributes—protocol, source port, and destination port—and one numerical attribute, which is the number of transactions in the corresponding minute. In our experiments, the three categorical features are used as the side information feature, and the numerical feature is used as the imputation feature. Following Dou et al. (2020), the transactions from non-IoT devices are considered as anomalies.

The UNB dataset [3] is a benchmark dataset produced by Sharafaldin et al. (2019) for DDoS (Mirkovic & Reiher, 2004) network attack detection. In our experiments, we use network transactions collected on March 11, 2019. The ground-truth labels of attack and benign sequences are available. This dataset includes 15,990 DDoS attack sequences and 15,989 benign sequences. In our experiments, the attack sequences are considered anomalies. Each data point in a sequence contains 11 attributes. We use the two numerical attributes for imputation and the remaining nine attributes as side information.

The SecVendor dataset, provided by a cybersecurity vendor, contains 102,073 real-world network transaction sequences from 1,835 users. A sequence consists of a series of HTTP requests from a user to a hostname, where a data point represents an HTTP request. The sequences are labeled as either normal or abnormal by the vendor's security experts. The labelled abnormal transaction sequences are malicious ones, e.g. command&control servers which get ransomeware deployed (Almashhadani et al., 2019). Each data point in a sequence includes 18 attributes. In our study, we use the three numerical attributes for imputation and the remaining 15 categorical attributes as side information.

The data points in all the time-series are ordered chronologically. We construct the training and test datasets using the earliest 90% and remaining 10% of each time-series, respectively. As mentioned in Section 4.4, the anomaly scores are converted to binary labels, that is, "normal" and "abnormal", using a threshold. To facilitate this, we segment each time-series in the training dataset into two parts. The earliest 60% of the points in each time-series are used for training our SSSD model, and the remaining 40% are used to establish the anomaly score threshold for classification. For the DDoS dataset, we adopt the upsampling method as described in Doriguzzi-Corin et al. (2020) to balance the positive and negative labels. No such balancing is performed for the other two datasets. All models take a fixed length time-series as input, which is set to 64 in our experiments.

| Dataset | # Sequences | Imputation Features | Side Information Features | Sequence Length | % Anomaly |
|---|---|---|---|---|---|
| IoT | 30,722 | 1 | 2 | 64 | 38.28% |
| DDoS | 31,979 | 2 | 9 | 64 | 50% |
| SecVendor | 102,073 | 3 | 15 | 64 | 62.22% |

Table 1: Some statistics of the datasets

### 5.1.2 BASELINES

We compare the performance of our proposed method with three representative generative baselines.

1. **LSTM-CVAE (Wang & Wan, 2019)** is the state-of-the-art time-series anomaly detection model. It is an LSTM-based CVAE model, where LSTM is employed as the encoder. The initial state of the decoder is the combination of the side information variables and the final state of the encoder.
2. **SSSD (Alcaraz & Strodthoff, 2022)** is the state-of-the-art generative model for time-series data. Leveraging structured state space models for its internal architecture (Gu et al., 2022), SSSD is good at capturing long-term dependencies within time-series data.
3. **UNet1D-DDPM (Huang et al., 2023)** is a type of DDPM (Ho et al., 2020b) model employing 1D convolutional U-Net Ronneberger et al. (2015) as the base denoising model for handling sequence data. Lugmayr et al. (2022) has shown that the performance of such models can be enhanced by incorporating conditional information. In our experiments, we leverage the side information features as the conditional element to guide the output generation.

All methods employ reconstruction errors as their anomaly scores and use the same approach for determining the anomaly score threshold to classify inputs as anomalous or not. LSTM-CVAE

---

[3]https://www.unb.ca/cic/datasets/ddos-2019.html

employs a single-layer LSTM encoder with a 128-dimensional hidden state and a five-layer decoder with layer dimensions of 512, 1024, 2045, and 2048, respectively. The SSSD and UNet1D-DDPM methods have the same network structure as the ones proposed in their original papers. We use the published Python codes of SSSD ⟨https://github.com/AI4HealthUOL/SSSD⟩. The remaining algorithms are implemented in Python. All experiments are conducted on a Google Cloud Platform virtual machine with an A100 GPU, 12 vCPUs, and 85GB RAM.

## 5.2 Efficacy and Efficiency Study

First, we evaluate the efficacy of all models in detecting anomalous time-series. The evaluation metrics, namely, Precision, Recall, F1 Score, and Area Under the Precision-Recall Curve are reported in Table 2.

| Method | IoT | | | | DDoS | | | | SecVendor | | | |
|--------|------|------|------|------|------|------|------|------|------|------|------|------|
|        | P | R | F1 | AUC | P | R | F1 | AUC | P | R | F1 | AUC |
| CVAE | 0.57 | 0.27 | 0.37 | 0.58 | 0.54 | **0.67** | 0.59 | 0.53 | 0.23 | 0.94 | 0.37 | **0.49** |
| DDPM | **0.65** | 0.36 | 0.46 | 0.56 | 0.51 | 0.55 | 0.53 | 0.49 | **0.26** | **0.99** | **0.41** | 0.44 |
| SSSD | 0.58 | 0.44 | 0.50 | 0.60 | 0.55 | 0.61 | 0.58 | 0.58 | 0.23 | 0.62 | 0.33 | 0.41 |
| Ours | 0.60 | **0.55** | **0.57** | **0.63** | **0.57** | 0.62 | **0.59** | **0.63** | **0.26** | **0.99** | **0.41** | 0.44 |

Table 2: "P", "R", "F1", and "AUC" are short for precision, recall, F1 score, and area under the precision-recall curve, respectively. "CVAE" and "DDPM" are short for "LSTM-CVAE" and "UNet1D-DDPM", respectively. The best records are in bold.

As shown in Table 2, our method consistently outperforms the baseline models, with two exceptions: the precision on the IoT dataset and the recall on the DDoS dataset. In the two cases, our method ranks second. LSTM-CVAE underperforms our proposed method due to its higher sensitivity to dataset noise compared to diffusion models, compromising its anomaly detection accuracy. Additionally, the known vanishing and exploding gradient issues in LSTM models hinder the learning of long-term dependencies, causing instability (Qin et al., 2023).

Our method consistently outperforms SSSD. The observation can be explained as follows. The reconstruction error can generally be partitioned into two segments: one stemming from the inherently anomalous nature of the input time-series, and the other arising from noise introduced by an inadequate model. Compared to SSSD, our method can effectively leverage the side information to better capture the input data distribution (Demonstrated in Table 3). Therefore, by minimizing the noise component in the reconstruction error, our method generates a more accurate anomaly score, resulting in superior performance for time-series anomaly detection.

Our method shows a more substantial improvement in performance over UNet1D-DDPM than over SSSD on the IoT and DDoS datasets. This suggests that the architecture of the underlying generative model is a more critical factor for performance enhancement than the incorporation of side information. However, the above observation does not hold on the SecVendor dataset, where UNet1D-DDPM has the same performance as our method. This discrepancy is likely due to the richer set of side information features in the SecVendor dataset, which appears to offset the advantages of a better model architecture.

To further study the effectiveness of our proposed method in learning the input dataset's distribution, we evaluate its imputation performance against the baseline methods. Table 3 shows the average imputation errors for all methods, including the reconstruction errors for LSTM-CVAE. Our method consistently outperforms the baselines, showing lower errors across all datasets. Next, we conduct a case study to illustrate the superior performance of our proposed method in generating high-quality time-series. Limited by space, we only show one case study on an IoT time-series, more case studies can be found in Appendix A.1. In the experiment, we randomly mask a continuous span of 32 data points in a time-series. The unmasked segments are fed into the models to reconstruct the masked segment. Figure 2 shows both the original time-series and the reconstructed time-series segment. Obviously, our method can reconstruct the masked segment much more accurately than baselines.

Finally, we study the efficiency of our proposed method. In all experiments, we use batch_size=512. The LSTM-CVAE converges quickly after 15 epochs on all datasets. In contrast, the other three

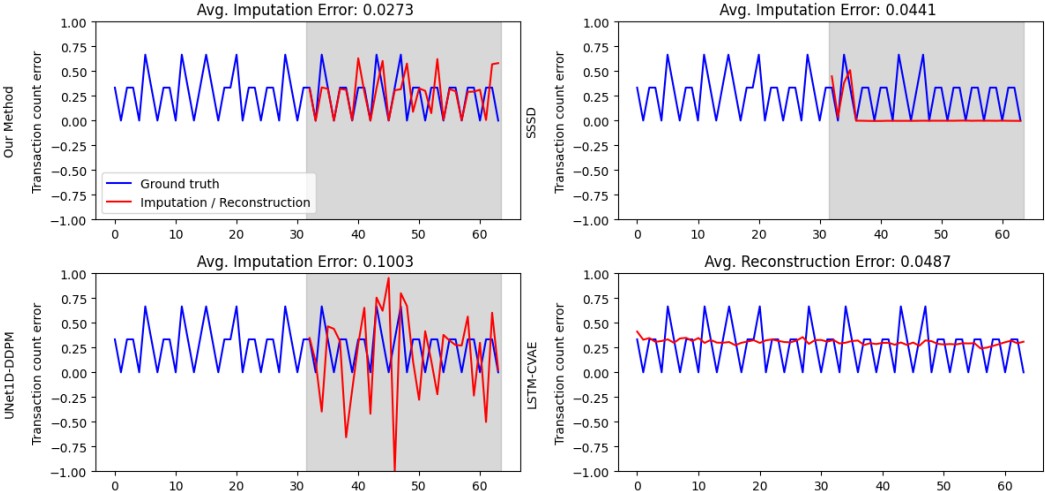

Figure 2: Imputation results for an IoT time-series using various methods. The blue curve represents the original time-series, while the red curve is generated by the respective generative model. Time steps that are masked are highlighted in gray.

| Dataset | LSTM-CVAE | UNet1D-DDPM | SSSD | Our Method |
|---------|-----------|-------------|------|------------|
| SecVendor | 0.45 | 0.27 | 0.24 | **0.22** |
| IOT | 0.09 | 0.15 | 0.06 | **0.04** |
| DDoS | 0.0010 | 0.5074 | 0.0015 | **0.0004** |

Table 3: Average imputation errors of all methods. The best records are in bold.

methods require approximately 60 epochs for convergence. However, as shown in Table 2, LSTM-CVAE cannot effectively detect anomalous time-series. While UNet1D-DDPM has a shorter training time per epoch, it suffers from the longest inference time per batch. LSTM-CVAE's performance varies significantly depending on the dataset. On the SecVendor dataset, despite its quicker training time, it performs worse than both the SSSD and our method at the inference stage. This suggests that LSTM-CVAE struggles with datasets having high-dimensional side information features. Both our method and SSSD perform consistently well across all datasets. However, as indicated in Table 2, our method outperforms SSSD in anomalous time-series detection with a clear margin.

| Method | Training Time (seconds per epoch) | | | Inference (seconds per batch) | | |
|--------|------|------|-----------|------|------|-----------|
| | IoT | DDoS | SecVendor | IoT | DDoS | SecVendor |
| LSTM-CVAE | 1 | 2 | 218 | 0.03 | 0.03 | 18 |
| UNet1D-DDPM | 0.72 | 0.79 | 17 | 78 | 78 | 155 |
| SSSD | 29 | 38 | 67 | 14 | 13 | 29 |
| Our method | 29 | 40 | 68 | 15 | 13 | 30 |

Table 4: Training and inference time for each model.

# 6 CONCLUSION

This paper puts forward a novel time-series anomaly detection method based on diffusion models. To the best of our knowledge, this is a pioneering study about leveraging time-series imputation for anomaly detection. In our proposed method, we exploit a couple of advantageous design choices, i.e. employing SSSD model and integrating side information. Based on our empirical results from three different datasets/tasks, the novel method demostrates promising capabilities and robustness.

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

# A    APPENDIX

## A.1    CASE STUDY

In this section, more representative examples of imputation results in three datasets are listed.

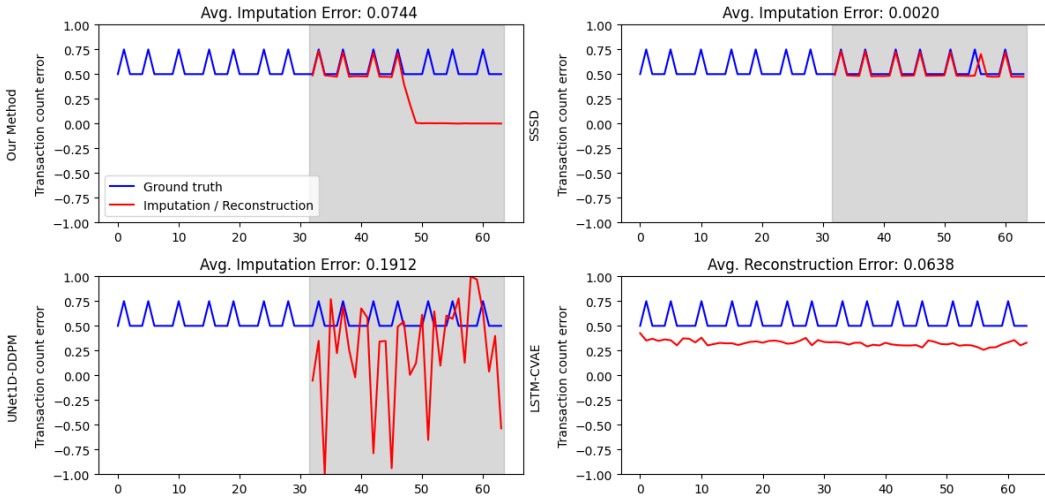

Figure 3: IoT Time Series Imputation example 1. The blue curve represents the original time-series, while the red curve is generated by the respective generative model. Time steps that are masked are highlighted in gray.

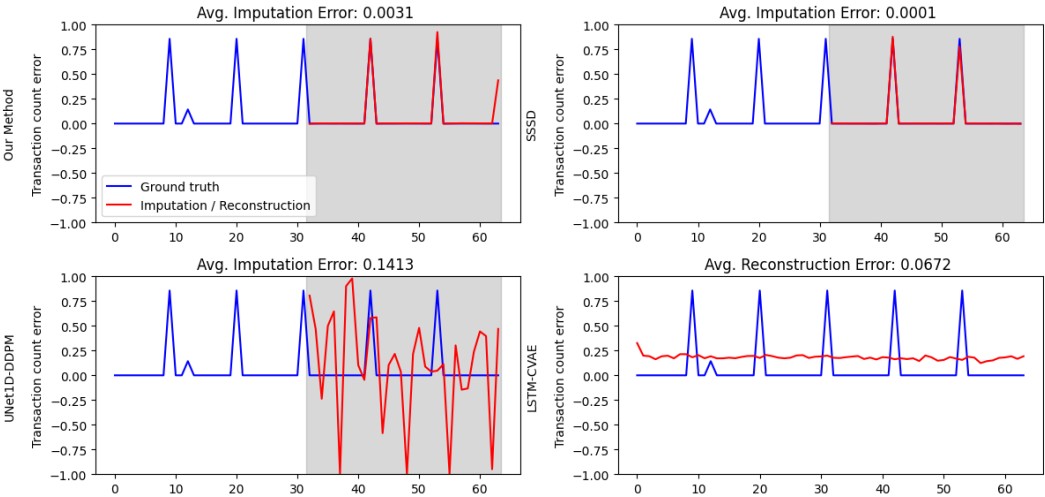

Figure 4: IoT Time Series Imputation example 2. The blue curve represents the original time-series, while the red curve is generated by the respective generative model. Time steps that are masked are highlighted in gray.

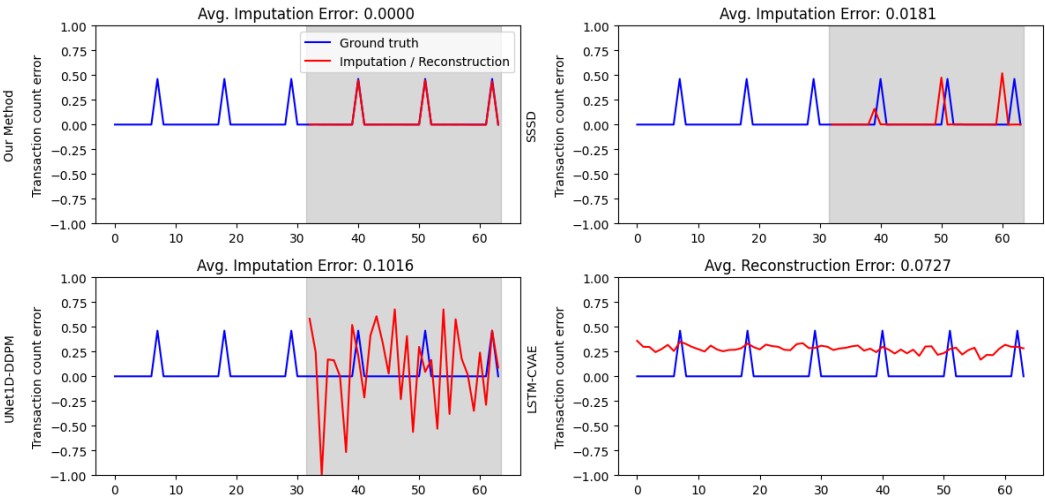

Figure 5: IoT Time Series Imputation example 3. The blue curve represents the original time-series, while the red curve is generated by the respective generative model. Time steps that are masked are highlighted in gray.

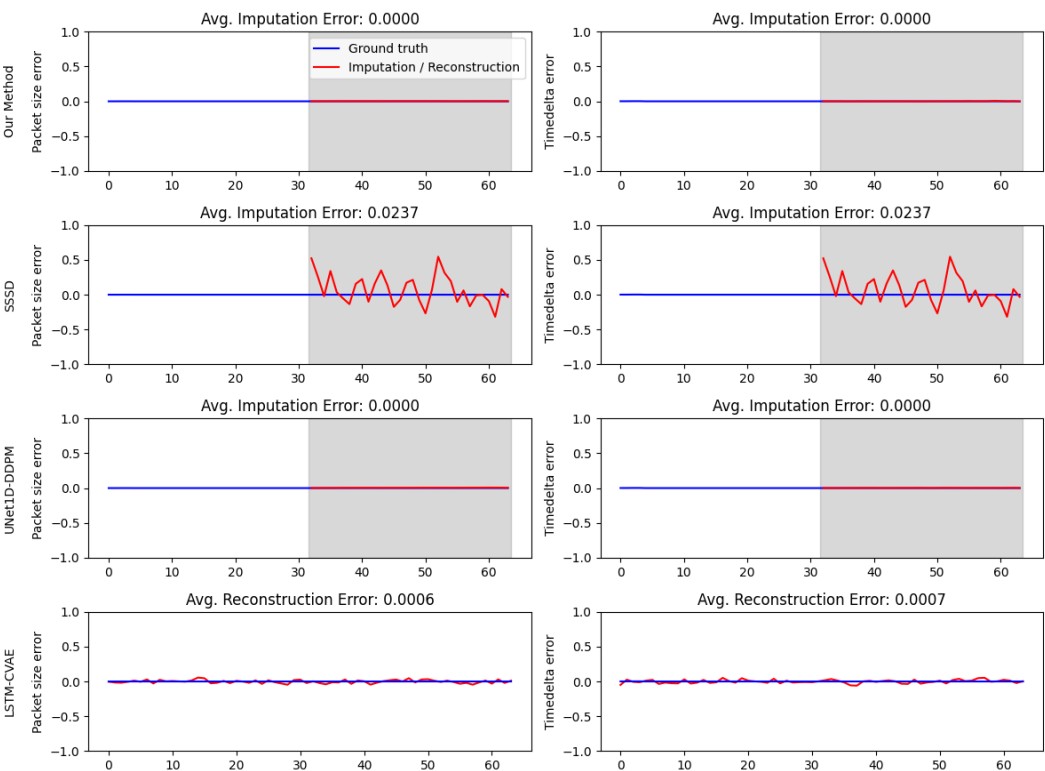

Figure 6: DDoS Time Series Imputation example 1. The blue curve represents the original time-series, while the red curve is generated by the respective generative model. Time steps that are masked are highlighted in gray.

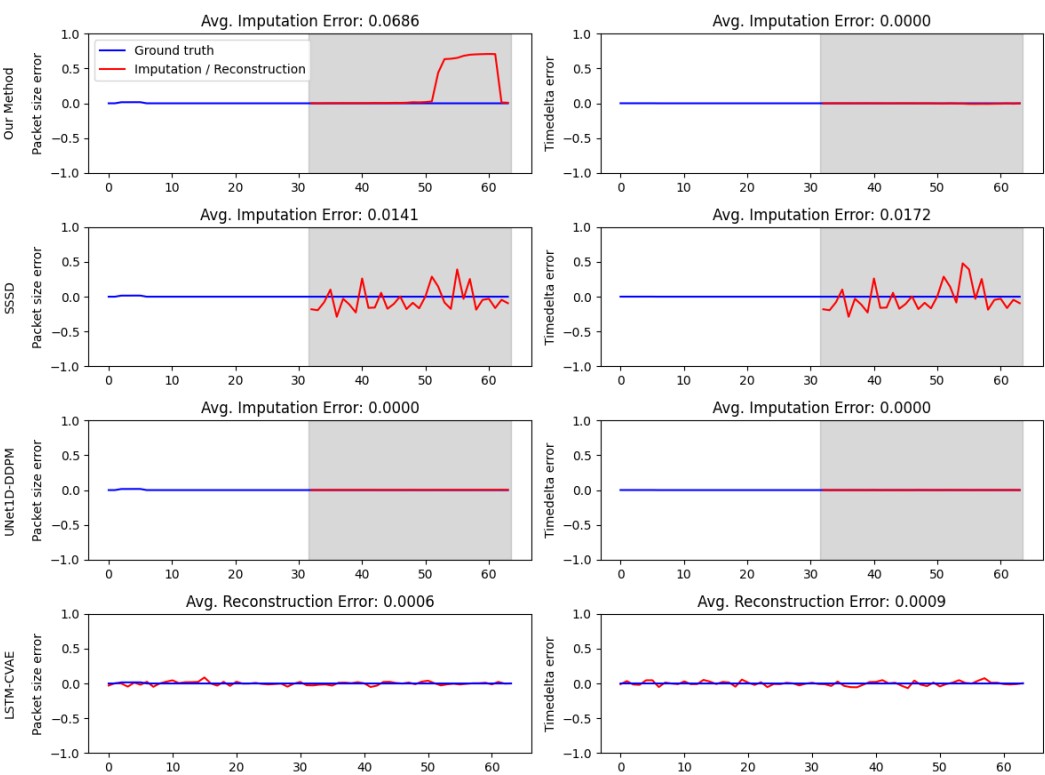

Figure 7: DDoS Time Series Imputation example 2. The blue curve represents the original time-series, while the red curve is generated by the respective generative model. Time steps that are masked are highlighted in gray.

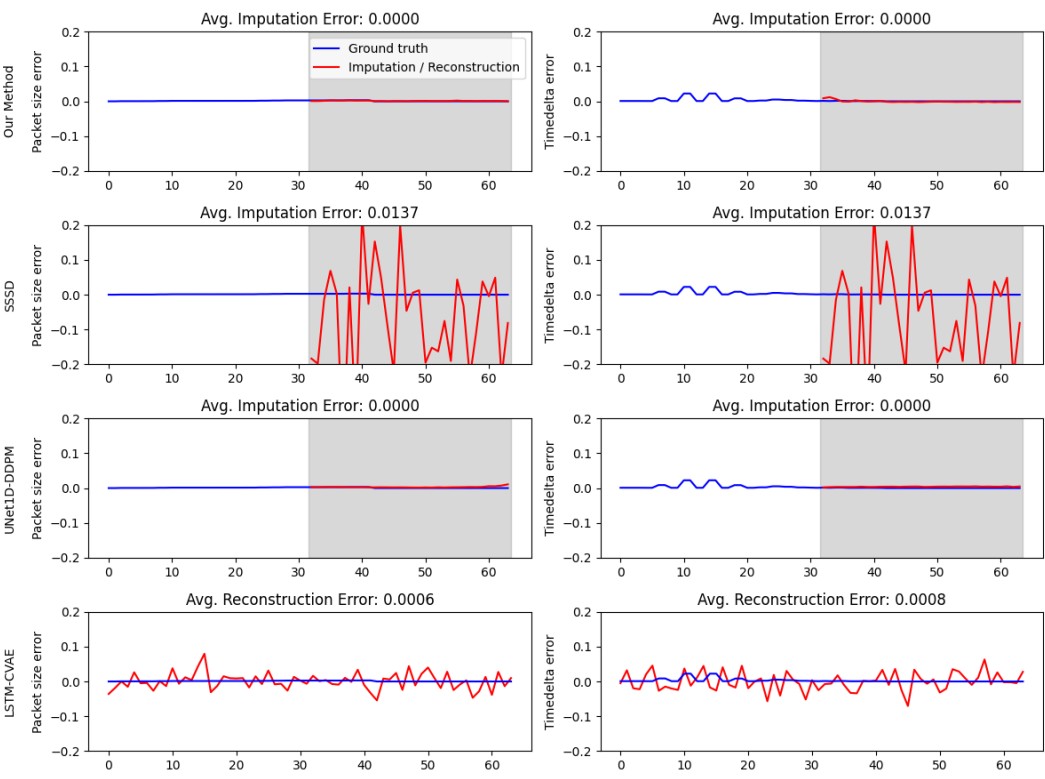

Figure 8: DDoS Time Series Imputation example 3. The blue curve represents the original time-series, while the red curve is generated by the respective generative model. Time steps that are masked are highlighted in gray.

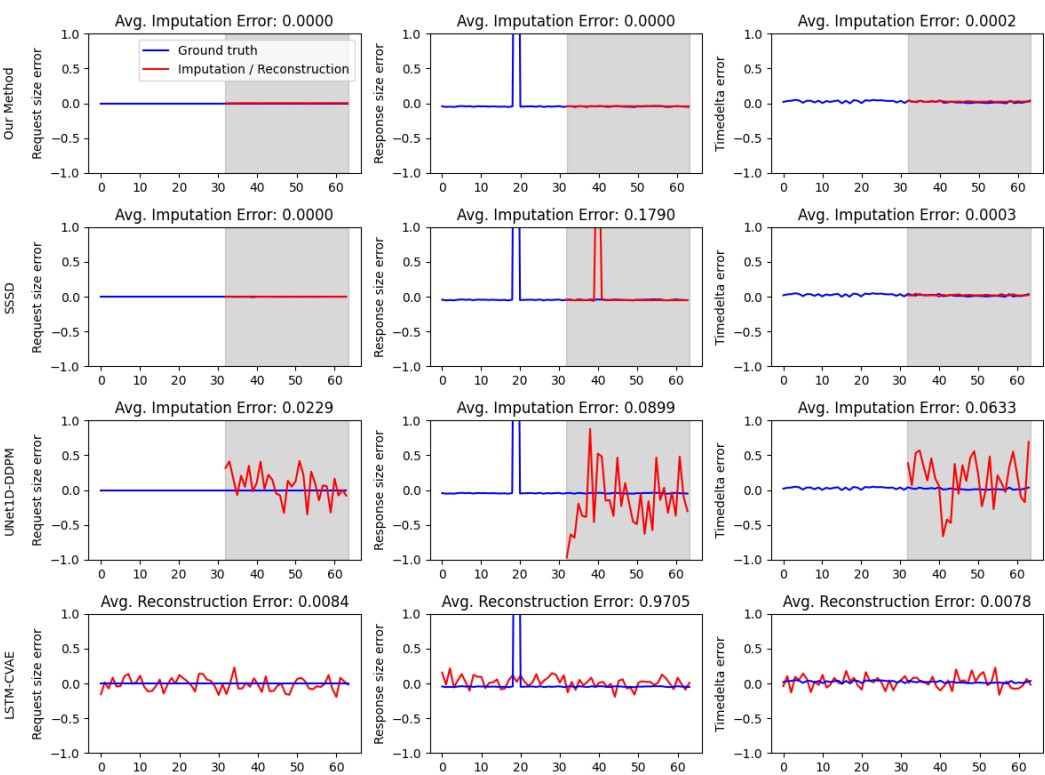

Figure 9: SecVendor Time Series Imputation example 1. The blue curve represents the original time-series, while the red curve is generated by the respective generative model. Time steps that are masked are highlighted in gray.

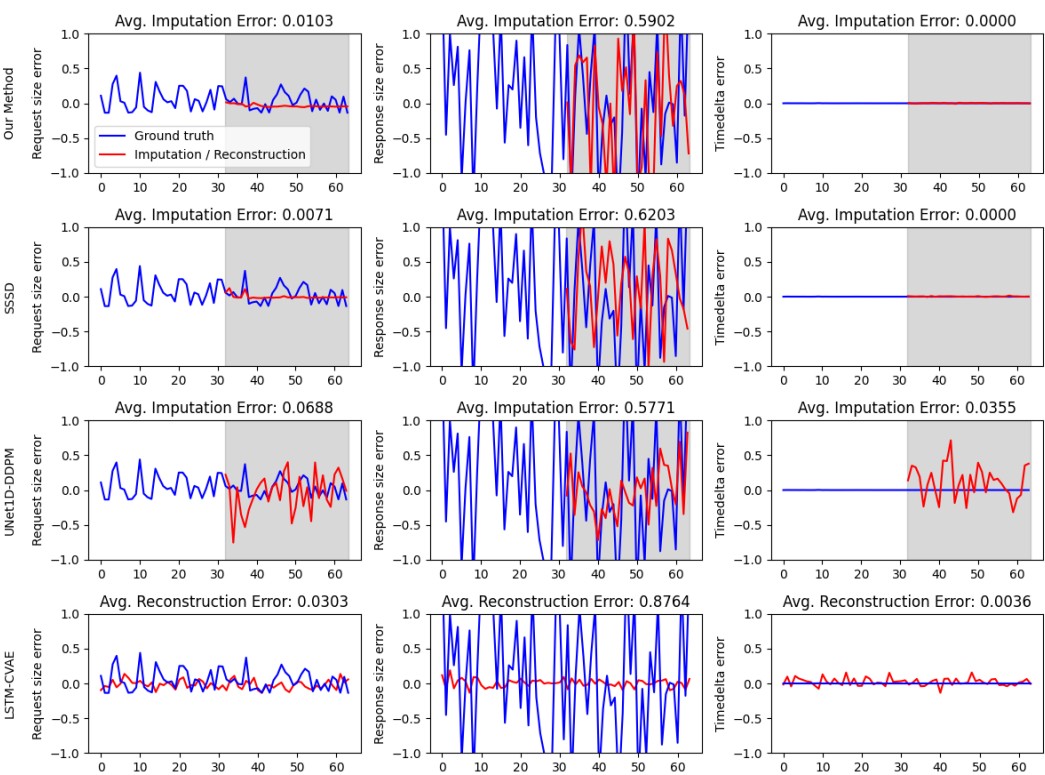

Figure 10: SecVendor Time Series Imputation example 2. The blue curve represents the original time-series, while the red curve is generated by the respective generative model. Time steps that are masked are highlighted in gray.

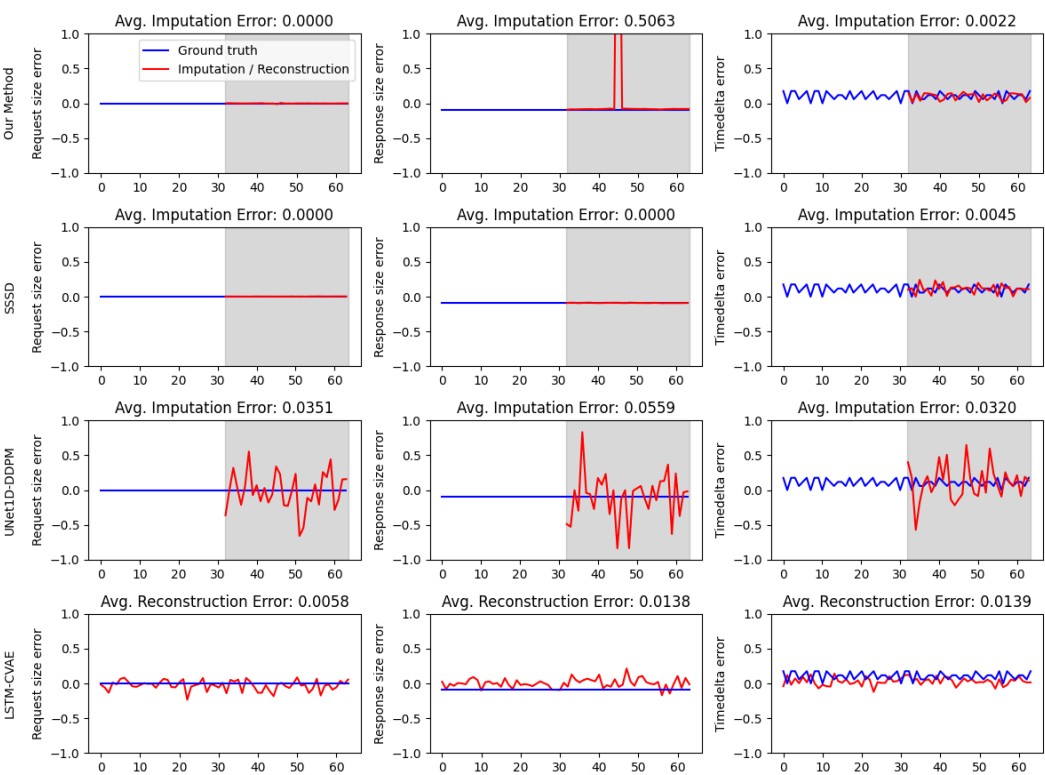

Figure 11: SecVendor Time Series Imputation example 3. The blue curve represents the original time-series, while the red curve is generated by the respective generative model. Time steps that are masked are highlighted in gray.

