# OpenReview forum: "Leveraging Heterogeneous Side Information via Diffusion Models for Time-series Anomaly Detection"
_ICLR.cc/2024/Conference — ICLR 2024 Conference Withdrawn Submission_

### Official Review · Reviewer_fEiX · 2023-11-01

**Soundness:** 3 good
**Presentation:** 3 good
**Contribution:** 1 poor
**Rating:** 3
**Confidence:** 3

**Summary:**

The paper presents a demonstration of how diffusion models can be used to leverage side information in order to solve time series anomaly detection (TSAD)  problem. The overall idea is to train a time series diffusion model on the training data (samples from in-distribution) and then assess the anomaly score of a test time series based on how well the trained diffusion model could reconstruct it using the imputed version of the given test time series along with the available domain specific side information. the paper uses Structured State Space Diffusion (SSSD) model that uses state space models as its internal model architecture. The  proposed approach is evaluated against three baselines including SSSD model without side information,  LSTM-CVAE model (a transformer-based conditioned VAE model), and UNet-1DDPM (a variant of Denoising Diffusion PM). are compared on two public benchmark datasets for TSAD problem and one production dataset from a prestigious cybersecurity vendor. The results indicate superiority of proposed approach over the three baselines on F1 and AUC metric.

**Strengths:**

Strengths:
1. The paper illustrates a nice demonstration of  SSSD model's ability to leverage side information in the context of time series anomaly detection problem.
2. The proposed approach performs better in comparison to the aforementioned three baselines on F1-score and AUC evaluation metrics.
3. The paper is well written and mostly easy to follow.

**Weaknesses:**

Weaknesses:
1. Technical contributions of this paper are quite limited in my opinion. All they are demonstrating is an application of SSSD model on a time series anomaly detection problem and how its performance improves with leveraging side information in comparison to vanilla SSSD model as well as two other baselines (LSTM-based and DDPM-based). While SSSD is a state-of-the-art model that has been applied to time series anomaly detection problems. Similarly, the idea of using side information in various diffusion models is also not new. Given these two facts, the technical contributions are quite modest.
2. The proposed approach is only evaluated against three baselines from time series generative models which is otherwise quite rich and have several other models that are potentially capable of leveraging  domain-specific side information but were not included in the evaluation study.
3. The related work (and evaluation) lacks coverage on the rich literature that exclusively focus on time-series anomaly detection problems.  In addition, the related work does not include some of the recent works on time series anomaly detection using diffusion models, e.g. [1] and [2].

References:
1. Imputation-based Time-Series Anomaly Detection with Conditional Weight-Incremental Diffusion Models
2. DDMT: Denoising Diffusion Mask Transformer Models for Multivariate Time Series Anomaly Detection

**Questions:**

1. As per Figure 1, it seems like we are feeding imputed time series to the diffusion model. What do you mean by imputed time series? Is it a time series with some values randomly erased and then replaced with imputed values? Also I wonder why can't we directly feed raw time series to the diffusion model? Is the imputed time series a specific requirement of SSSD model? Or do SSSD models perform better with imputed time series? In any case, I will recommend authors to provide a brief discussion (or a relevant reference) justifying this step.

2. In section 2.1 Eq(2), should LHS be p(x_0, x_1, ..., x_{t-1}, x_{T})? Basically, it should not be a conditional distribution in LHS since RHS includes p(x_T)?

3. Questions on Figure 1:
a. What are the two colors indicating in time series plots?

b. Both good reconstruction and bad construction look exactly similar to me. What's the difference?

c. Why do we need imputed time series? why can't we work with original time series?

---

### Official Review · Reviewer_Xkjb · 2023-11-04

**Soundness:** 3 good
**Presentation:** 2 fair
**Contribution:** 2 fair
**Rating:** 5
**Confidence:** 4

**Summary:**

This work proposes an anomaly detection method using the diffusion model. The authors claim that this is the first work of using the diffusion model (DM) for anomaly detection. The model takes an extra input of a K-dimensional vector as side information, which is used as part of the input of the DM.

While DM provides a generative model (= probability distribution), the authors use non-probabilistic and non-information theoretic anomaly metric, which is essentially the same as the t-score.

**Strengths:**

- The use of the DM for anomaly detection, which can be viewed as a relatively new application of the DM.
- Provide a minor modification by taking an extra input as side information

**Weaknesses:**

- Almost all the description is about the DM. The authors' technical contribution is not clear.
- Significant inconsistency between DM as a generative model and the definition of the anomaly score.

**Questions:**

Please justify the definition of the anomaly score in light of existing anomaly detection works that uses information theoretic or generative metrics such as negative log-loss.

---

### Official Review · Reviewer_nb7N · 2023-11-05

**Soundness:** 3 good
**Presentation:** 2 fair
**Contribution:** 3 good
**Rating:** 5
**Confidence:** 4

**Summary:**

The authors propose a time-series anomaly detection model that can incorporate side information. The method uses time-series diffusion models as its backbone but modifies it to include the side information. The anomaly scores are computed based on the reconstruction error from the diffusion model. The higher the scores, the more anomalous the data is. The authors then demonstrate the benefit of the models in several datasets.

**Strengths:**

Strengths:
- Time series anomaly detection is an important area to study, as there are many applications of the methods in this area.
- The use of diffusion models for anomaly detection is interesting.
- The model can incorporate side information into the detection.

**Weaknesses:**

I have a few concerns and questions regarding the paper:
1) The paper lacks a detailed description of how the diffusion models (SSSD) work in the mentioned setting, time series input with random masking, and side information. Section 2.1 only explained the generic diffusion model. It is not clear how to incorporate the masking and side information into the process.
2) The algorithm section (section 4.4) only describes the prediction stage using a trained SSSD. However, it does not explain anything about how to train the SSSD model, particularly related to the integration of side information into the SSSD model.
3) Questions about experiment setup:
     - For the construction of training and test sets, does the split based on the order in the sequence, i.e., each sequence is split into two where the first 90% of the data points are used as training and the second 10% are used as the test? Similarly, does the similar splitting is also used in the thresholding setting?
    - In the real-world data, the length of time series data are usually varied. Can the model accommodate that? As in the experiments, all the lengths are fixed.
    - In many anomaly detection literature, the anomaly cases are usually far less than normal cases. However, in the experiments' datasets, the anomaly cases can even be larger than normal cases. Could the authors explain more about this setup?
     - As the proposed model also uses SSSD as the trained model, could the author explain more about the difference between SSSD and the proposed model in the experiments?

**Questions:**

Please answer my questions in the previous section.